# Fast Prediction for Resistance Spot Welding Deformation Using Inherent Strain Method and Nugget Model

**DOI:** 10.3390/ma14237180

**Published:** 2021-11-25

**Authors:** Takeshi Chino, Atsushi Kunugi, Toshikazu Kawashima, Goro Watanabe, Cao Can, Ninshu Ma

**Affiliations:** 1JSOL Corp., Tokyo 104-0053, Japan; kunugi.atsushi@jsol.co.jp (A.K.); kawashima.toshikazu@jsol.co.jp (T.K.); 2Geely Automobile Research Institute (Ningbo) Co., Ltd., Ningbo 315336, China; g.watanabe@geely.com (G.W.); caocan@geely.com (C.C.); 3Joining and Welding Research Institute, Osaka University, Osaka 567-0047, Japan

**Keywords:** resistance spot welding, welding deformation, inherent strain method, inherent deformation, nugget model, finite element method

## Abstract

In a car body, there exist thousands of resistance spot welds, which may induce large deformation during the manufacturing process. Therefore, it is expected that automotive industries will develop a method and a computing system for the fast and simple prediction of its deformation. Although the inherent strain method has been used for the fast prediction of arc welding deformation, it has not been applied to resistance spot welding so far. Additionally, the electrical-thermal-mechanical coupling analysis for the deformation induced by resistance spot welding is complicated and much more time-consuming. Therefore, in this study, a nugget model of the resistance spot weld has been developed, and the inherent strain method is extended for use in the fast prediction of resistance spot welding deformation. In addition, the deformation of a vehicle part with 23 resistance spot welds was efficiently predicted within around 90 min using the inherent strain method, displaying good accuracy compared with the measurement.

## 1. Introduction

Resistance spot welding (RSW) is the primary joining method in the automotive manufacturing process due to its high productivity and low cost [1]. A typical vehicle can contain thousands of spot welds [2,3]. Many studies on resistance spot welds have been carried out with regard to their crash safety assessment under the impact loading [4,5,6,7] and fatigue strength improvement [8,9,10]. On the other hand, such a large number of resistance spot welds may induce severe deformation during the assembling process. Therefore, it is strongly anticipated that the automotive industries will develop a fast and simple method for predicting RSW induced deformation.

RSW is a complicated process that involves mechanical, thermal, electrical, and metallurgical phenomena [11]. The primary numerical approach to RSW deformation is based on the electrical-thermal-mechanical coupling analyzing method [12,13,14]. Many coupling analyses have been conducted to investigate weld-nugget formation [15,16,17]. However, the coupling analysis requires (i) a lengthy computing time to predict the structure deformation due to a lot of spot welds, (ii) temperature-dependent physical properties, and (iii) accurate modeling of the contact between welding electrodes and the welded materials to transfer the electric current.

Besides the coupling analysis method, the inherent strain method has been successfully applied to predict arc-welding induced deformation [18,19,20,21,22] and is recognized as a fast and simple method [23,24,25]. More specifically, the proposed inherent deformation concept for the database establishment of arc welding, using the measured deformation in a simple test specimen [26,27,28,29], makes the inherent strain method more practical for welding deformation prediction at the design stage, i.e., before production. The inherent deformation described the final thermal deformation due to welding, which is given as the integral of the inherent strain existing in the plastic deformation zone including the melted zone and the heat-affected zone(HAZ) [30]. The assembly deformation, a result of welding, can be efficiently predicted by applying the inherent deformation level sequentially to each weld line as shown in Figure 1.

However, as of yet, the inherent strain method has not been applied for RSW. Therefore, it is necessary to extend the inherent strain method to RSW and to develop an analytical system of RSW deformation for practical applications.

There are two main steps in the RSW process, the electrode pressure step and the welding step. It is necessary to classify the inherent deformation modes induced by the electrode pressure step and welding step so that spot weld deformation induced by complicated phenomena can be expressed using a simple methodology as opposed to the conventionally inherent strain method [30].

In this paper, the inherent deformation modes that exist in a resistance-spot weld are specified and their corresponding magnitude parameters are defined. The inherent deformation parameters are identified by measuring the deformation of a single spot weld specimen. In addition, the deformation of a vehicle part with 23 resistance spot welds was predicted using the inherent strain method and analysis system.

## 2. Primary Deformation Mechanism in Resistance Spot Welding

In the automotive industry, high accuracy is always expected in assembling automotive parts. Since the phenomena in the RSW process are complicated, many factors influence on the welding quality. From the perspective of the deformation generation mechanism, the following four influencing factors have to be considered and numerically modeled.
Initial geometrical imperfection of work-piece;Gap between work-pieces before welding;Electrode pressure (e.g., Indentation in the work-piece surface);Inherent strain due to welding thermal cycle.

First, the initial geometrical imperfection of the work-piece, depending on accuracy of the stamping process, can be considered in FE modeling, based on the deformed geometry after stamping.

Second, work-pieces with a gap that has been induced in previous manufacturing processes, such as stamping, are generally closed as a result of electrode pressure, as shown in Figure 2 before the welding current flows. The process of the gap closure can be reproduced, similar to the root gap in the case of an inherent deformation analysis of arc welding [31].

The third and fourth items are both related to the creation of the weld nugget and can be modeled by extending the inherent deformation concept for RSW.

## 3. Inherent Deformation for RSW

### 3.1. Concept of the Inherent Deformation Analysis

The inherent strain method, which was first proposed by Ueda et.al [32], is the basis of the inherent deformation analysis. Although complex phenomena are produced by welding, the total strain in the welding process can be expressed by Equation (1) [33].
(1)$$\varepsilon =\varepsilon^{e}+\varepsilon^{p}+\varepsilon^{T}+\varepsilon^{cr}+\varepsilon^{tr}$$
where $\varepsilon^{e}$ is elastic strain, $\varepsilon^{p}$ is plastic strain, $\varepsilon^{T}$ is thermal strain, $\varepsilon^{cr}$ is creep strain and $\varepsilon^{tr}$ is phase transformation strain. Equation (1) can be rearranged as follows:(2)$$\varepsilon -\varepsilon^{e}=\varepsilon^{p}+\varepsilon^{T}+\varepsilon^{cr}+\varepsilon^{tr}=\varepsilon^{*}$$
where $\varepsilon^{*}$ is the inherent strain. Equation (2) means that the deformation and the residual stress are caused by the inherent strain. Therefore, the overall deformation during the welding process can be predicted by elastic analysis using the inherent strain distribution as the initial strain.

The inherent deformation analysis [30] is widely used as it is a simpler way to use the inherent strain method efficiently. Instead of distributing inherent strains in the melted zone and the HAZ, a database of inherent deformation can be easily created and managed by classifying each deformation mode as an inherent deformation. The entire deformation of the welding process can be predicted by considering inherent deformation as uniform initial strain in the welding elements.

### 3.2. Strain Distribution at the Weld Nugget

To study the basic behavior of the nugget generation of RSW, an electrical-thermal-mechanical coupling analysis was performed using Ansys LS-DYNA. Figure 3 shows a single spot weld FE model for the coupling analysis.

Figure 4a displays the distribution of the z-directional strain on the cross-section of the weld nugget. The z-strain is found at the center of the weld nugget, so the deformation is concentrated at the center. Additionally, the distribution of the x-directional strain in the z-direction is minimal at the center of the weld nugget and maximal at the edge, induced by electrode pressure and thermal expansion, respectively. The x-directional strain, which is indicated in Figure 4b, is concentrated at the edge of the nugget. Radial-shrinkage deformation and the out-of-plane bending-by-bending moment are induced by the x-directional strain.

### 3.3. Deformation Modes of RSW and Nugget Element

From the results of the previous section, deformation induced by the spot welding process is found to be concentrated inside the weld nugget. As in an conventional inherent deformation analysis, by classifying the inherent deformation from the strain distribution of the weld nugget, it is possible to predict the overall deformation of the welding process by considering the inherent deformation as the initial strain to the elements in the weld nugget area.

To reproduce deformations induced by the RSW process, three types of inherent deformation modes, shown in Figure 5, are classified. The deformation modes of in-plane shrinkage, out-of-plane shrinkage, and out-of-plane bending are defined for the upper and lower plates, respectively. The magnitude of these deformations can be configured using distributed inherent strain $\varepsilon_{ri}{}^{*}\left(r,\;z\right)$ and $\varepsilon_{zi}{}^{*}\left(r,\;z\right)$ with the following Equations (3)–(5) in which the t is the plate thickness and subscript i denotes the lower plate if it is at a value of 1, and the upper plate if it is at a value of 2. Even a three-sheet spot weld can be expressed by an extension of these equations.
(3)$$S_{ri}^{*}=\frac{1}{t}\iint \varepsilon_{ri}{}^{*}\left(r,\;z\right)dzdr,\;\left(i=1,\;2\right)$$
(4)$$S_{ti}^{*}=\frac{1}{t}\iint \varepsilon_{zi}^{*}\left(r,z\right)dzdr,\;\left(i=1,\;2\right)$$
(5)$$R_{ri}^{*}=\frac{12}{t^{3}}\iint z\varepsilon_{ri}^{*}\left(r,z\right)dzdr,\;\left(i=1,\;2\right)$$

Here, $S_{ti}^{*}$ represents the change in the thickness at the weld point, which includes indentation induced by electrode pressure. The modeling of the electrodes and its enforcement are not required because inherent deformation reproduces the indentation.

The three deformation modes exist in RSW, and their magnitudes can be described via inherent strain for a simple and fast prediction of the deformation. Here, the nugget model was implemented at the nugget area as shown in Figure 6. The shape of the developed nugget model is an octagon, meshed by 8 shell elements that are positioned with a pitch of 45 degrees along the circumferential direction. This nugget model can reproduce the real nugget shape with better accuracy. Elements of the nugget model between parts are modeled as interface elements [30], so complete bonding between parts is generated after welding.

## 4. Deformation Calculation Procedures

Figure 7 shows the flowchart of inherent deformation analysis. The procedure is identical to the conventional inherent deformation method for arc welding. First of all, inherent deformation data needs to be prepared. As for RSW, a single spot weld specimen is used for identifying inherent deformation based on the following Equation (6):(6)$$\left\{D^{*}\right\}={\left[H^{*}\right]}^{-1}\left\{D^{m}\right\}$$
where, $\left\{D^{*}\right\}$ and $\left\{D^{m}\right\}$ are inherent deformation vectors of ($S_{ri}^{*},\;S_{zi}^{*},R_{ri}^{*}$) and measured visible welding deformation vectors of ($S_{ri}^{m},\;S_{zi}^{m},R_{ri}^{m}$), respectively. [$H^{*}]$ is the elastic response matrix. The *j*th column of [$H^{*}]$ is obtained from the results of the elastic analysis with the *j*th component of $\left\{D^{*}\right\}$ as 1 and the others as zero [28]. The magnitude of the three inherent deformation modes of the single spot-weld specimen was identified using the changes of the measured distance and angle between the measuring points located on the top and bottom surfaces as shown in Figure 8.

Then, the FE model was constructed by performing meshing, defining jig constraint or weld order based on the imported geometry data of the structures. Finally, an elastic finite element analysis was conducted using the inherent deformation data for RSW. At the analysis step, both the gap closing induced deformation and the inherent strain-induced deformation were calculated in sequence.

## 5. Application Example

### 5.1. Single Spot Weld and Identification of the Inherent Deformation

A single spot weld was conducted to obtain the inherent deformation data. The test specimen and the measuring points of its deformation are shown in Figure 8, and its materials are shown in Table 1. The materials used for the test specimen were the same as those used for vehicle parts, aide from the fact that the upper part was not plated. In order to identify the deformation of a single spot weld, the welded shape was measured by the geometry-measuring device, Keyence VR5200, before and after welding.

The data used for the identification of the inherent deformation was prepared by extracting the deformation from the scanned geometry at the measuring points shown in Figure 8. The FE model for the identification of inherent deformation and its evaluation points are shown in Figure 9. The points highlighted in orange correspond to the measuring points that are placed in the same position as the test specimen, shown in Figure 8, and the inherent deformations $S_{r}^{m}$, $S_{t}^{m}$, and $R_{r}^{m}$ at these points were evaluated to identify the magnitude of inherent deformations. For $S_{r}^{m}$ and $R_{r}^{m}$, the evaluation values for two directions are calculated for the upper and lower plates, respectively.

A comparison between the measured geometry and the reproduced geometry by the inherent strain method is shown in Figure 10. X- and Y-sections centered on the weld point are presented, and only the area overlapping 50 mm of the specimen is visualized. The difference between the reproduced geometry and the measured geometry is a result of the assumption of axisymmetric deformation in the inherent deformation analysis. The asymmetry in the actual weld, as seen in the measured case, is mainly affected by the way the electrode contacts the plate material and the accuracy of the initial shape of the plate. In order to level out such effects in the calculation of the inherent deformation analysis, the identification calculation was performed so that the axisymmetric deformations are matched. Therefore, although $S_{r}^{*}$ and $R_{r}^{*}$ have been computed in two directions, their average values are evaluated to identify the inherent deformation. Figure 11 shows the identified magnitude of inherent deformation parameters $S_{r}^{*}$, $S_{t}^{*}$, and $R_{r}^{*}$ corresponding to in-plane shrinkage, out-of-plane shrinkage, and out-of-plane bending, respectively, and these parameters were calculated from 18 measuring points on the top and bottom surfaces of the plate as shown in Figure 9. For $S_{r}^{*}$ and $R_{r}^{*}$, the x- and y-directions were evaluated, and the identification calculations were performed so that their average values agreed with the measured results. These deformation modes showed good agreement with the measured modes.

### 5.2. Welding Conditions of Vehicle Parts

Figure 12 shows the vehicle parts and welding points. The vehicle parts consist of two parts. Red numbers show welding orders. Table 1 and Table 2 show specs of each component and the weld conditions, respectively.

A lever-type clamp was used for the vehicle parts. The clamps were set by hand up to welding point 12, then the clamps were released so that it was supported by hand, and welding continued until welding point 23.

### 5.3. Finite Element Modeling

To develop the finite element model, the geometry of vehicle parts was measured by a 3D geometric scanner before welding. A finite element model consisting of shell elements was developed based on the scanned geometry, with initial geometrical imperfection induced by the stamping process. The model is shown in Figure 13. It has round 33,000 nodes and elements. In the inherent deformation analysis, Young’s modulus, Poisson’s ratio, and the mass density are 206,000 [MPa], 0.3, and 7.5 × 10^−9^ [ton/mm^3^], respectively, which are typical values of a steel plate at room temperature because only elastic analysis is performed. Geometrical nonlinearity is taken into account. Part-to-part and part-to-clamp contact is also taken into account, resulting in static and dynamic friction coefficients of 0.3 and 0.1, respectively. Clamps were placed with a margin of about 3 mm from the position specified by the CAD.

### 5.4. Results

#### 5.4.1. The Results Obtained by the Proposed Method

Figure 14 shows the welding deformation mode (a) predicted by the proposed inherent deformation method and (b) measured by a 3D scanner, respectively. The deformation analysis of 23 weld points was efficiently performed within around 90 min, using this method. Note that the electrical-thermal-mechanical coupling analysis of single-spot welding, shown in Figure 3, took around 60 min. It can be visually judged that the overall deformation mode was well predicted. To quantitatively validate the predicted deformation magnitudes at all measuring points shown in Figure 15, the bar graph of the welding deformation, shown in Figure 14, was used in Figure 16. We found that the predicted deformation at the evaluation points are generally in good agreement with the measured deformation. In particular, the level of deformation at measuring points 1, 7, 8, or 12, where welding is concentrated, are in good agreement, suggesting that the deformation induced by the welding process, such as electrode pressurization, can be reproduced.

#### 5.4.2. The Effect of the Nugget Model Distortion Due to Deformation

The nugget model is defined based on the initially designed geometry, so the nugget model may be distorted due to the welding assembling operations. The mechanism of the nugget-model distortion is shown in Figure 17. The interface element of the nugget model is defined by the initial geometry, and its nodes are shared with other parts. If parts are deformed or moved due to jig constraint or welding, interface elements of the nugget model are distorted. In the current implementation, the loading of the gap closing is applied to the center of the nugget model, as shown in Figure 18. Applying the load of the gap closing, based on this axial misalignment, can result in deformation that differs from the actual electrode pressurization.

From the result of predicted deformation shown in Figure 14a, it can be observed that severe deformation occurred especially at weld number 23, which is the final welding point. The time history of nugget-model distortion at the weld 23 is shown in Figure 19, presenting a distortion of more than 3 mm at the end of the weld process. It can be seen that distortion at weld 23 was caused by the jig constraint and nearby welding points. In order to verify the effect of this distortion on the predicted deformation results, an analysis for the modelled and manually updated mesh, at weld 23, was conducted. Based on the prediction results in Figure 19, the mesh has been adjusted to take into account the distortion using the procedure illustrated in Figure 20.

Figure 21 shows a comparison of the measured and predicted deformation of the upper part. Figure 21a presents the computed deformation in cases where the coordinates of the nugget model were kept as the initial ones, shown in the previous section. Figure 21b presents the results for when the mesh of the nugget model is updated at weld 23, according to the procedure in Figure 20. It can be seen that the results of the model (b) are very close to the measured geometry at the upper end of the model, indicating that the predicted result was improved near the area the mesh updated. Local deformation was greatly improved by modifying the nugget-model distortion, and the difference of the deformation between model (a) and (b) was found to be substantially lower in other areas. An appropriate method by which to avoid problems caused by a distortion of the nugget model without manually mesh updating should be the subject of future work.

## 6. Conclusions

In this study, the concept of the conventional inherent deformation analysis was extended to predict the assembly deformation of spot welds. The main conclusions are summarized as follows:(1)Based on the deformation generation mechanism in resistance spot welding, three types of inherent deformation modes, in-plane shrinkage, out-of-plane shrinkage, and out-of-plane bending were classified, and their magnitude is expressed by corresponding parameters.(2)The octagon nugget model was developed and meshed by 8 shell elements positioned with a pitch of 45 degrees along the circumferential direction.(3)The deformation that occurs in assembling the vehicle model with 23 resistance spot welds was calculated efficiently by the proposed method and validated by experiments. The calculated weld deformation agreed well with the measured deformation.(4)By updating the nugget mesh to avoid the extreme distortion of the nugget, the predicted results were found to be much closer to the measured results. Future studies should explore ways in which to avoid issues caused by a distortion of the nugget model.

## Figures and Tables

**Figure 1 materials-14-07180-f001:**
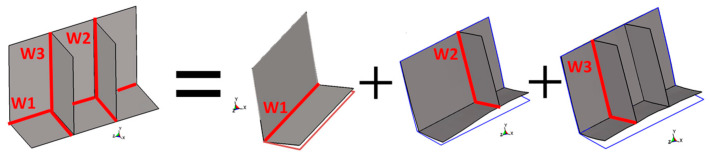
Analyzing sequence of arc welding induced deformation with the aid of inherent strain method.

**Figure 2 materials-14-07180-f002:**
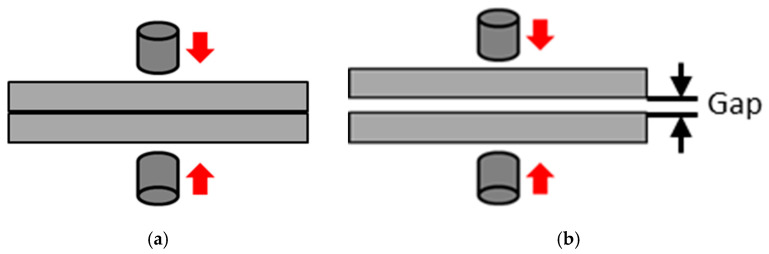
Gap between work-pieces. (**a**) Ideal; (**b**) In real.

**Figure 3 materials-14-07180-f003:**
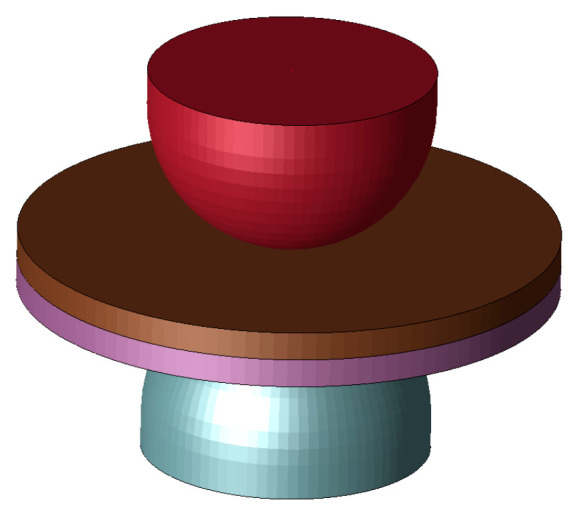
FE model for the process simulation.

**Figure 4 materials-14-07180-f004:**
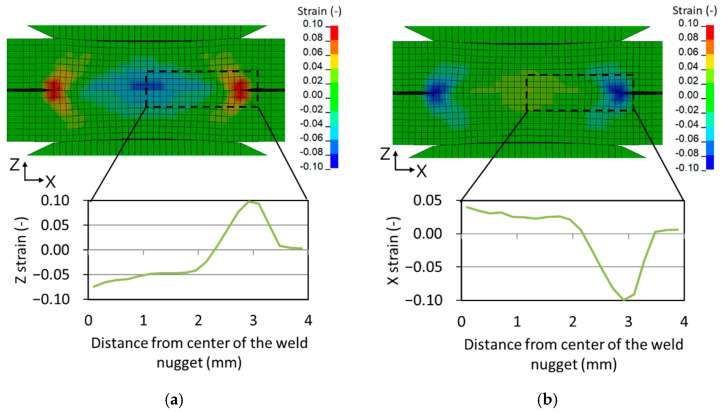
Strain distribution in the weld nugget predicted by process simulation. (**a**) The z-strain; (**b**) The x-strain.

**Figure 5 materials-14-07180-f005:**
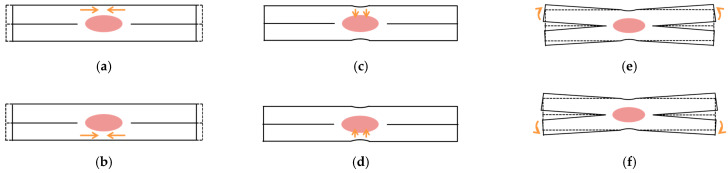
Deformation modes for RSW. (**a**) In-plane shrinkage (upper part); (**b**) In-plane shrinkage (lower part); (**c**) Out-of-plane shrinkage (upper part); (**d**) Out-of-plane shrinkage (lower part); (**e**) Out-of-plane bending (upper part); (**f**) Out-of-plane bending (lower part).

**Figure 6 materials-14-07180-f006:**
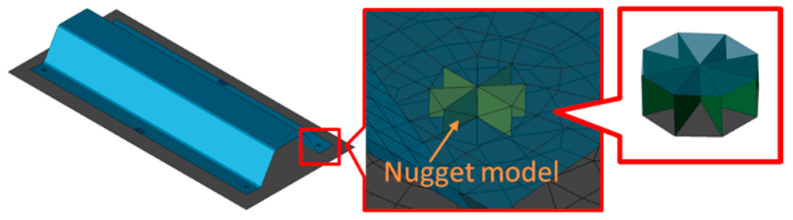
Nugget model.

**Figure 7 materials-14-07180-f007:**
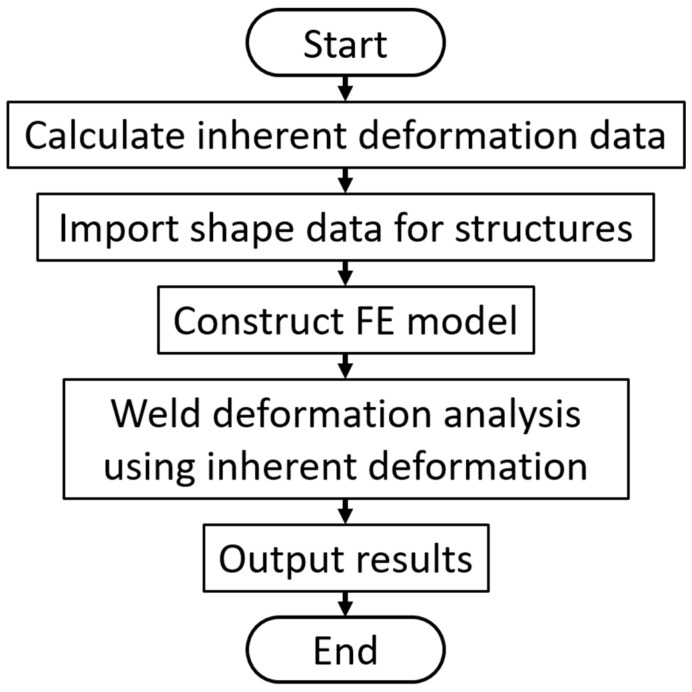
Flowchart of inherent deformation analysis.

**Figure 8 materials-14-07180-f008:**
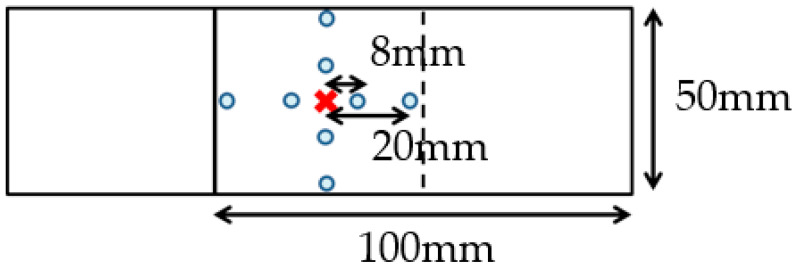
Single spot specimen and measuring points.

**Figure 9 materials-14-07180-f009:**
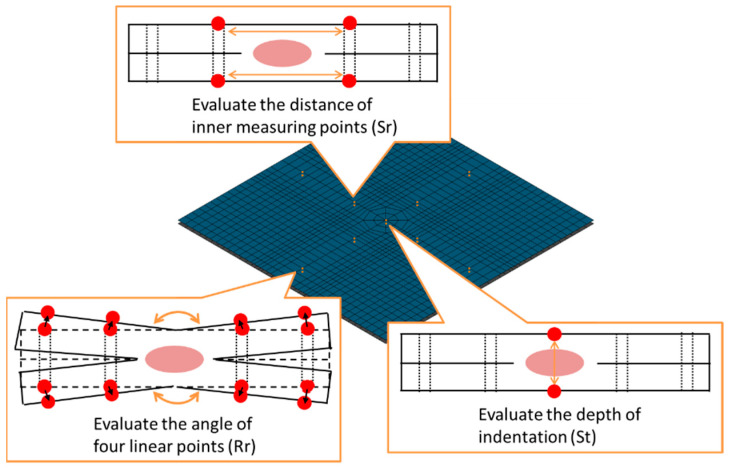
FE Model for identification of inherent deformation. The distance and angle for evaluating Sr, St, and Rr are computed at the measuring points highlighted in orange.

**Figure 10 materials-14-07180-f010:**
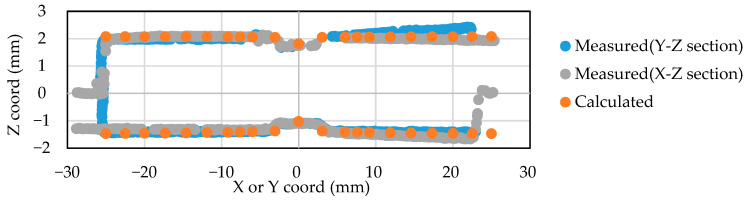
Comparison of the scanned geometry and the calculated one for identifying inherent deformation.

**Figure 11 materials-14-07180-f011:**
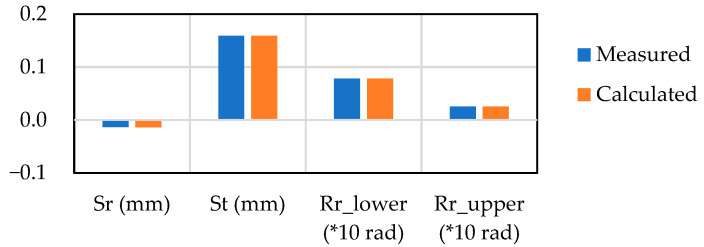
Fitting of the axisymmetric inherent deformation modes.

**Figure 12 materials-14-07180-f012:**
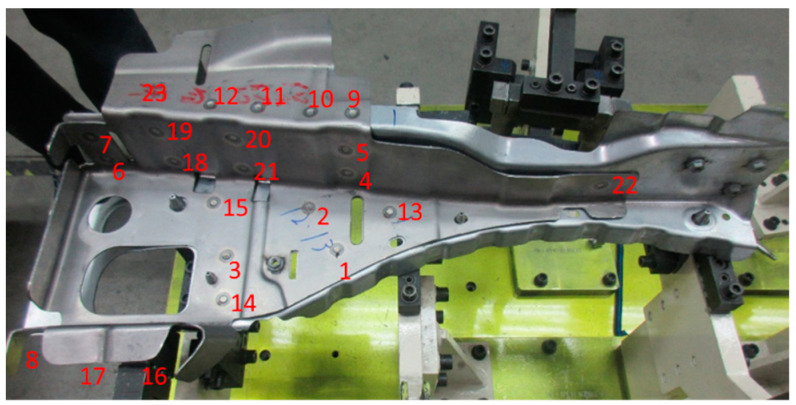
Welded vehicle parts. The red numbers indicate welding orders.

**Figure 13 materials-14-07180-f013:**
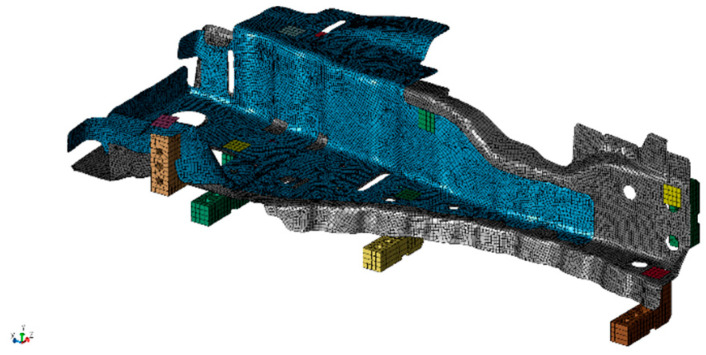
Finite element model for vehicle parts and clamps. Blue is the upper part, gray is the lower part, and the others are clamps.

**Figure 14 materials-14-07180-f014:**
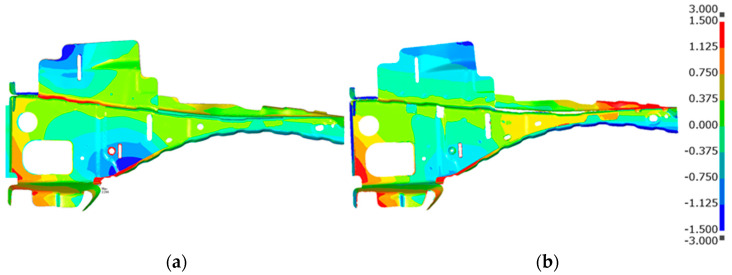
Welding deformation (mm). (**a**) Predicted; (**b**) Measured.

**Figure 15 materials-14-07180-f015:**
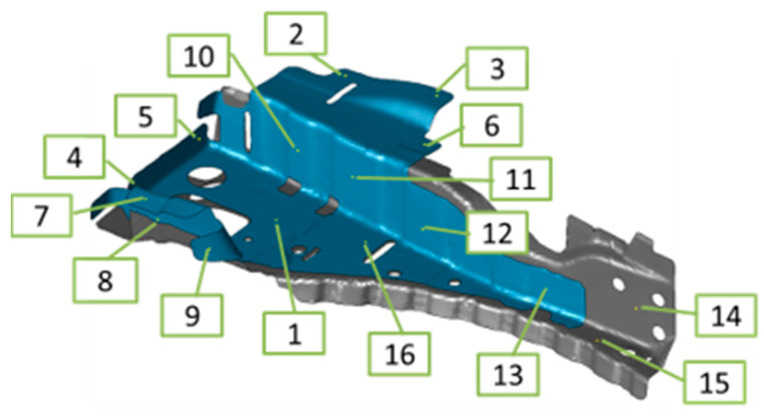
Measuring points.

**Figure 16 materials-14-07180-f016:**
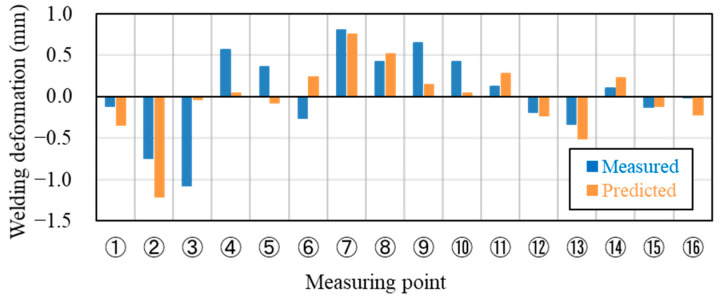
Bar graph of welding deformation at the measuring points.

**Figure 17 materials-14-07180-f017:**
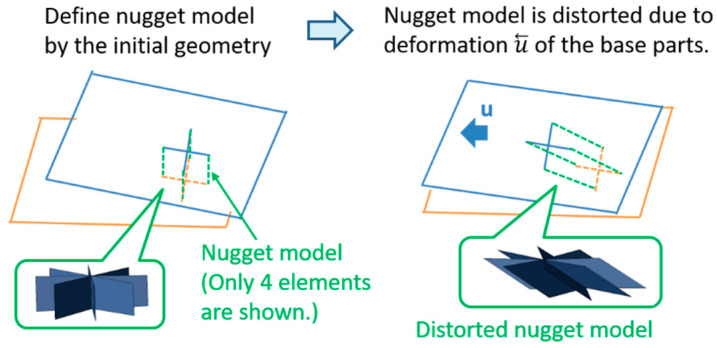
Distortion of the nugget model.

**Figure 18 materials-14-07180-f018:**
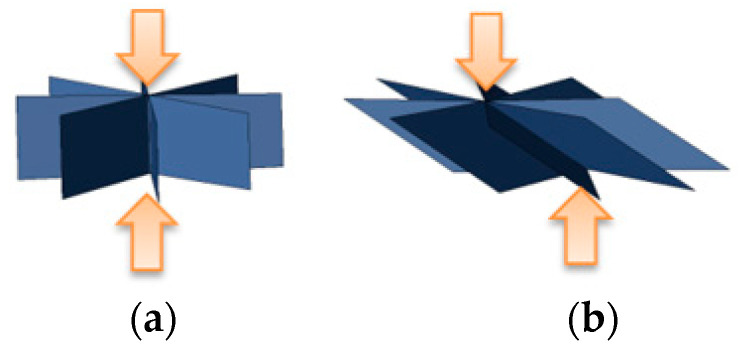
Differences in loading conditions of gap closing. (**a**) Ideal condition; (**b**) Distorted condition.

**Figure 19 materials-14-07180-f019:**
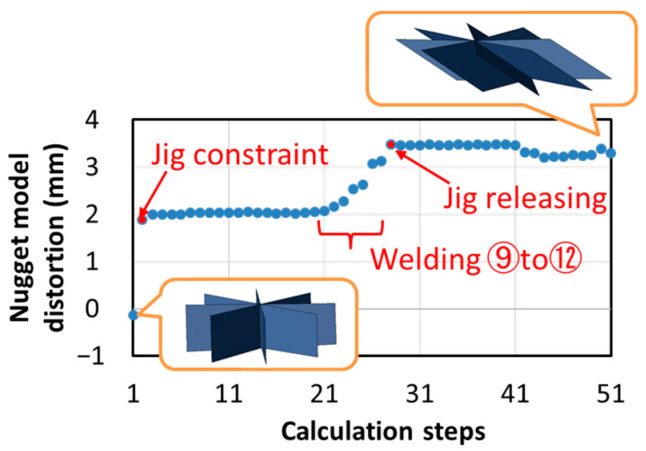
Time history of nugget model distortion at welding point 23.

**Figure 20 materials-14-07180-f020:**
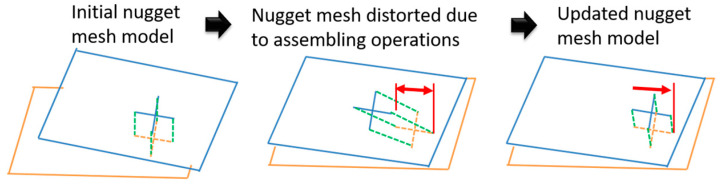
Nugget mesh model update for accurate prediction.

**Figure 21 materials-14-07180-f021:**
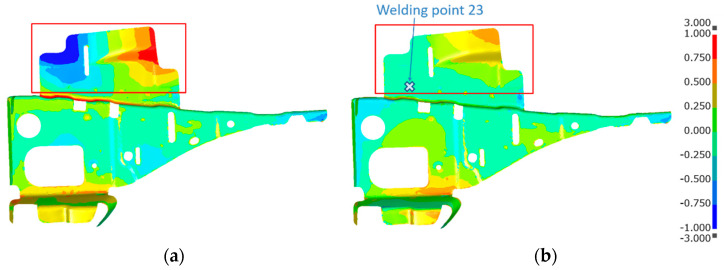
Comparison of measured and predicted geometry of the upper part. (**a**) Initial nugget mesh without update; (**b**) Nugget mesh at the weld 23 with update.

**Table 1 materials-14-07180-t001:** Spec of components for vehicle parts.

Component	Material	Thickness
Upper	DP590 grade (GI)	1.8 (mm)
Lower	DP590 grade (GI)	1.2 (mm)

**Table 2 materials-14-07180-t002:** Welding conditions.

Electrode Type	Electrode Tip Diameter	Electrode Force	Pre-Pulse		Main-Pulse		Hold Time
			Welding Current	Welding Time	Welding Current	Welding Time	
Truncated	6.0 (mm)	3.0 (kN)	12.0 (kA)	30 (ms)	8.5 (kA)	400 (ms)	600 (ms)

## Data Availability

The raw/processed data required to reproduce these findings cannot be shared at this time due to technical or time limitations.

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
