# Peer review of "Fast Prediction for Resistance Spot Welding Deformation Using Inherent Strain Method and Nugget Model"

_materials, 2021, doi:10.3390/ma14237180_

Round 1

Reviewer 1 Report

Dear Authors of the “Fast prediction for resistance spot welding deformation using inherent strain method and nugget model”, I find the article interesting and scientifically correct. Unfortunately, in my opinion, there are some fault that, should be corrected and supplemented:

- please provide a description of the numerical modeling method,

- the information presented in Figure 9 can be replaced by a single sentence in the text. After this change the  figure 8 can be enlarged and more visible.

- I would also like to pay attention to the tables 1 and 2. In my opinion, the work can be redrafted and one of them can be removed or combined,

- Fig. 14 will be difficult to read for the reader. It is impossible to unequivocally define the values presented on it. Please also place the error values with which these parameters were determined.

- Conclusion - requires a short description. In my opinion, a listing of the authors' achievements can be given as a description. However, in points, provide what are statements.

- Literature is too poor (only 17 position). Additionally, only one publication from the last 4 years. This have to be corrected

Reviewer 2 Report

The manuscript addresses the numerical simulation of deformation of the sheet stack during resistance spot welding deformation through an inherent strain method and nugget model.

The results of the manuscript could be interesting. However, some considerations are unclear or missing. I suggest the author revising the manuscript according to the following comments:

- The inherent strain method should be described in more detail in the Introduction paragraph to clarify which are the characteristics of this method, how it differs from more traditional FEM analyses, and why it can provide faster computations during FEM analysis.

- Replace the sentence “…electrical-thermal-structural coupling analysis…” in “…electrical-thermal-mechanical coupling analysis…".

- It is not clear to which FEM simulation the images of Figure 3a and 3b are referred to. In table 1, it is written that FEM simulation has involved 1.8 and 1.2 mm sheets. However, from Figure 3 it does not seem to refer to such sheets (i.e.,. the simulated sheets have the same thickness). Clarify this point.

- Clarify the meaning of the terms of equation 4. This is not so clear from what is written “Where, [?∗], [?∗] and [??] are inherent deformation vector of (??∗,??∗,??∗), elastic response matrix and measured visible welding deformation [12]”

- the authors should provide a Table reporting the electrical, thermal, and mechanical properties of the metal sheets at varying temperatures that were used for setting up the numerical simulations.

- How has the melting of the weld nugget managed in the numerical simulation? And the Microstructural phase transformations in the HAZ? Clarify. This is important for modeling the welded joint and the ensuing deformations (paragraph 3.2).

- The authors state that “The deformation analysis of 23 weld points was efficiently performed within about 90 minutes using this developed method”. By comparison, they should point out the time that should be required for a traditional FEM analysis.

- It does not clear why the authors refer to axisymmetric deformation in the inherent deformation analysis to justify the difference with the reproduced geometry (line 129-130). Could it also be due to the uncertainty related to the material properties (i.e. matching between real and simulated properties) or also to other factors?

- A photo of the real part could be provided along with fig. 10 and 11, or in the following fig. 12 and 13.

Finally, the main text has many misprints, like “may induces” etc. I suggest reviewing better the English grammar.

Reviewer 3 Report

The authors of the manuscript “Fast prediction for resistance spot welding deformation using inherent strain method and nugget model” tried to apply a promising approach to calculate deformation caused by resistance spot welding. This task is very interesting, and in case of a successful solution, it can expand the field of application of the method.

The overall impression is a very concise description of all procedures and results for a regular article. The descriptions of the figures are reduced to one or two sentences and a reference to publications, often in Japanese, which makes it very difficult to understand and evaluate the results obtained.

For example, a number of questions arise.

How the experimental points in fig.8 correlate with fig. 7?

The discrepancy for the top right side in fig. 8 is very significant. How, based on this result, can the method be applied to real nuggets in section 5.2?

Figure 9 shows the fitting results, but the text does not indicate what unknowns were fitted.

Section 5.2. Figs 10 and 11 demonstrate that the lower component is much thicker that the upper one, while Table 2 suggests otherwise.

Judging by the calculation results presented in fig. 3, the strain at a distance of more than the radius of the nugget is about zero. In this case, the main deformations are the result of pressure from clamps and electrodes. Then the question arises: does it make sense to use the inherent strain method for the RSW? How large is the contribution from inherent strain to the final deformation.

In general, I believe that the manuscript can be published after some additions to the text.

Other minor comments

What are the units of the scales in Fig.3? absolute or percent?

What do the red rectangles in fig. 3 and ovals in fig. 12 mean?

Match brackets and braces in Eq.(4) and in line 106.

Reviewer 4 Report

The paper describes a method to predict the defomations induced through resistance spot welding in a welding part using the inherent strain method. At first, the four different deformation mechanisms in resistance spot welding and the resulting different deformation modes are presented. Afterwards, the inherent deformation analysis is briefly described before an application in form of a single spot weld specimen is shown. In the following subchapters the inherent deformation analysis is extended to a welded vehicle part with 23 individual welding points. Following the results, a method to take the distortion of the welding nugget into account is introduced and successfully applied.

The English language of the paper needs to be revised as there are multiple grammatical errors, for example: incorrect conjugations of verbs or adverbs, (lines 11, 15, 67, 84…), missing “the” (lines 58, 60, 63, 64, 162,…) or missing pronouns (line 151).

The sentence  in line 66: “Second, work-pieces with the gap induced in previous manufacturing processes such as stamping, have to be assembled through enforcedly deforming or moving to the weldable position as shown in Figure 1.” is hard to understand as it is not clear to the reviewer which gap was already induced in manufacturing processes prior to the welding or how a insufficient described movement could help. Figure 1 doesn’t help to clarify this issue.

Furthermore, there are some formal errors (empty line 120 +127, missing reference in line 171, wrong letters in line 243). Chapter 6 is named discussion but only covers the nugget model distortion and remeshing of the model. Additionally it only has the single subchapter (6.1). The results from the single spot weld experiments show a good accuracy and hint that the method is working as expected . But the claim “that the predicted deformation at all evaluation points has a very good agreement with (the) measured one” is not supported by the presented data. The points 1, 2, 7, 8 and 11-15 in Figure 14 show an accurate conformity with the measured data, whereas the rest of the 16 measuring points do not match. Therefore, the drawn conclusion needs to be revised, especially since the error in the prediction is also partly explained in the following section by the nugget model distortion.

The paper covers an interesting topic. The developed method seems suitable for a time-efficient prediction of the deformations induced by resistance spot welding. The aforementioned shortcomings should be addressed before the paper can be published.

Round 2

Reviewer 3 Report

The manuscript can be recommended for publication in the revised version.

On line 238, in the sentence "... the identification calculations were performed so that their average values agreed with the calculation results", words "calculation results" should probably be replaced with "measured results".